# Influence of the Tree Decay Duration on Mechanical Stability of Norway Spruce Wood (*Picea abies* (L.) Karst.)

**Tomasz Jelonek [1],\*** , **Katarzyna Klimek [1]**, **Joanna Kopaczyk [1,2]**, **Marek Wieruszewski [3]** , **Magdalena Arasimowicz-Jelonek [2]**, **Arkadiusz Tomczak [1]** and **Witold Grzywiński [1]**

1   Department of Forest Utilization, Faculty of Forestry, Poznan University of Life Sciences, Wojska Polskiego 71A, 60-625 Poznan, Poland; katarzyna.klimek@up.poznan.pl (K.K.); joanna.kopaczyk@up.poznan.pl (J.K.); arkadiusz.tomczak@up.poznan.pl (A.T.); witold.grzywinski@up.poznan.pl (W.G.)

2   Department of Plant Ecophysiology, Faculty of Biology, Adam Mickiewicz University, Umultowska 89, 61-614 Poznan, Poland; m.arasimowicz-jelonek@uam.edu.pl

3   Department of Wood-Based Materials, Faculty of Wood Technology, Poznan University of Life Sciences, Wojska Polskiego 38/42, 60-627 Poznan, Poland; marek.wieruszewski@up.poznan.pl

\*   Correspondence: tomasz.jelonek@up.poznan.pl

**Abstract:** Wood properties have an influence on the safety around the tree itself as well as on actual possibilities of using wood. The article focuses on the wood properties of the Norway spruce (*Picea abies* (L.) Karst.) in reference to the time since the tree has decayed. The study was conducted among mature tree stands of spruce in Białowieża Forest, where over the last 10 years there has been a weakening of spruce tree stands due to water deficiency which has contributed to the gradation of the European spruce bark beetle (*Ips typographus*). The study focused on spruce wood of living and healthy specimens as well as the wood of standing trees which has decayed between one and five years before the sample was collected. The findings indicate a gradual decrease in wood properties as time passed since the physiological decay of the tree. Significant differences in the decrease of mechanical wood properties have been observed in trees which had been decayed for 3 years and they should be considered life and health hazard for people and animals.

**Keywords:** Bialowieża Forest; Norway spruce; modulus of elasticity; hazardous trees; tree decay

## 1. Introduction

Climate change is an invariable phenomenon which is caused both by anthropogenic factors as well as natural processes connected with volcanic activity and sun variability [1,2]. Regardless of the causes, the ramifications of climate change affect forests. Over the last decades, there has been an exponential increase in the number of new tree diseases appearing all over the world. There have also been diseases whose origin is complex and the abiotic factors, connected with climate change, should be taken into account [3]. Furthermore, climate changes can also be linked with the expansion of infectious diseases and their vectors, as well as immunity decrease of tress leading to susceptibility to diseases [2,4,5]. The process of tree stands' decay occurring at European Plain is a phenomenon which has been observed for a couple of decades. As a result of climate change, there has been a gradual withdrawal of coniferous species from the northern part of the continent.

In Europe there has been concern over spruce preservation, which undergoes gradual weakening, and in some parts decay, together with damages caused by wind and the gradation of the beetle (*Ips typographus*). These factors and spruce forests management became a subject of discussion

and academic scrutiny [6,7]. The problem touches spruce trees located in Białowieża Forest, which was inscribed on the World Heritage List in 1979. At the time, a little under 5000 ha of Białowieża National Park was inscribed on the UNESCO list. In 1992, the area was extended by the Belarussian part of Białowieża Forest, which then constituted a single cross-border facility, and in 2014, the World Heritage List encompassed the entire Polish part of Białowieża Forest. The dominating species in the area is the Norway spruce (*Picea abies* (L.) Karst.), which is one of the most important coniferous tree species in Europe and due to its ecologic and economic nature has become a subject of interest for ecologists and foresters.

The climate models, which are based on simulations, predict that European forests will be exposed to more frequent and longer droughts as well as higher temperatures [8]. The adaptation process to the changing conditions, in which the species grow, requires a long-lasting genetic (breeding) selection. The spruce, which grows better on damp areas and is not able to adapt to the changing climate quickly, decays and, according to some prognoses, within specified habitats may be superseded by species with better temperature tolerance [9]. However, the results of the study on the subject are not unambiguous. According to Briceno-Elizondo et al. [10] climate changes can significantly influence the formation of pine forests and their immunity. However, Hartl-Meier et al. [11] believes that the pace of pine growth in the Northern Alps did not change substantially due to climate changes in the 20th century, which could indicate that climate changes are not the only factors for tree stand decay.

The study conducted by Jyske et al. [12] indicate that the spruce is able to quickly return to physiological balance once the droughts have subsided. Hence, despite the noticeable influence of short-lasting droughts on the growth of spruce, in the long-term it probably will not have an influence on the quality of the wood. It should also be remembered that with such analyses it is difficult to differentiate single factors which shape a given feature; that is why the final wood properties are usually a resultant of numerous circumstances and stimuli influencing the tree growth. The changing environment, including climate, anthropogenic changes, and stress factors can have an impact on not only the tree growth and its physiological processes, but also on physical and mechanical properties of wood which have further consequences. In Wimmer and Grabner's example [13] the ratio of latewood increases as the amount of precipitation in the summer months increases, because the above average sum of precipitation causes earlier inhibition in earlywood formation, which results in a prolonged period of latewood formation. The increased sum of precipitation has a direct impact on wood density, because latewood is characterised by a higher density than earlywood due to the smaller diameter of tracheids, and also theoretically on the mechanical wood properties. The decay of spruce forest in Europe has been observed and scrutinised since the end of 1970s [14]. Over the last few years, a wide-spreading decay of forests on large areas of Central and Eastern Europe has been noticed. The forests seem to be attacked by a perplexing and lethal combination of air pollutants, NOx and heavy metals [15] together with natural stresses, including climate factors and pathogens [15,16]. Furthermore, air pollution in this area has a significant influence on the trees, which contributed to their weakening and decay that have been examined. Sulphur dioxide produced mainly by coal power plants and chemical industry has caused a considerable decay of spruce forests in the Czech Republic [17].

The study of historical factors impacting structural disorder of spruce tree stands in Central Europe conducted by Brůna et al. [7] indicate the age of the tree stand as one of the crucial factors. At the same time, the age of the tree stand was important for the formation of susceptibility to damage, it revealed the role of the forest's age structure in determining the scale of damages caused by wind and gradation of the beetle. The study of natural regeneration of spruce tree stands conducted by Svoboda et al. [6] reveals that eliminating dead trees from decaying spruce forests in semi-natural forests is harmful as it lowers capability for natural regeneration. On the other hand, leaving the decayed and standing trees in the forest or in urban areas generates the risk of harming people, animals, and property damage [18,19].

Nowadays tree managing methods, such as removing, nurturing and mechanical resistance assessment are mainly based on visual evaluation or human experience [20]. The majority of models refer to the risk of one damaged tree or the entire tree stand appearing, but not all factors are taken into consideration which is understandable due to their number and considerable variability. One of the key factors, apart from external forces, morphological architecture of the tree, or the properties of the tree stand, are the values of mechanical resistance of the xylem. These, however, are characterised by a natural variability (changeability) within a single species, and they can be influenced by numerous factors such as fungal pathogens. Honkaniemi et al. [21] described the influence of *Heterobasidion* on pine strength and the risk of damages due to wind in tree stands infected with Heterobasidion annosum.

The value of mechanical resistance of wood is largely conditioned by ultrastructuring anatomy and is characterised by a considerable variability even within a single tree, especially in radial direction [22], which is closely connected with the changeable angle of inclination of the microphibrils in the tracheid walls of latewood. In spruce there is a strong and negative correlation between the angle of the microphibrils and modulus of elasticity [20,23,24].

The research into the properties of pine and spruce in Finland and France conducted by Verkasalo and Leban [25] revealed the mean values for spruce—12,872 [MPa] [modulus of elasticity] MOE in Finland and 10,035 [MPa] MOE in France. In the case of wood resistance to static bending MOR, the mean values were respectively 82.3 [MPa] and 58.9 [MPa]. The results obtained by Verkasalo and Leban [25] are similar to the values of modulus of elasticity for spruce in Poland obtained by Bacher and Krzosek [26]. The mean value of modulus of elasticity at bending (MOE) for spruce, according to the authors, was 11,991 [MPa]. Following the research presented by [22] the modulus of elasticity at bending for spruce depended on geographical location, thus in France the mean was 11,200 [MPa] (at 12% humidity), in Poland 10,400–11,300 [MPa], in Romania 9100–9600 [MPa], in Sweden 10,400–11,200 [MPa], and Slovenia 11,000–12,000 [MPa].

Lavers [27] and McLean [22] reveal that in the case of spruce wood deprived of defects the mean resistance to bending strength is between 59 $Nmm^{-2}$ and 67 $Nmm^{-2}$, however in wood with defects it can drop even to 30 $Nmm^{-2}$. Another very important mechanical property of wood is modulus of elasticity, which according to Lavers (1983) in a healthy spruce wood is on average 8100 $Nmm^{-2}$, however, according to McLean [22] it is 6800 $Nmm^{-2}$. The modulus of elasticity, similarly to resistance to static bending strength, depends on numerous properties of the xylem and the presence of defects and anomalies in wood. It can be assumed that in the case of decaying trees there are changes in the xylem not visible to the naked eye, which significantly influence the wood properties and, at the same time, its stability and safety around it.

The aim of the article is to analyse the basic mechanical properties in assessing wood stability of standing trees at various stages of decay. The main assumption in the study was the occurrence of a natural depreciation of wood in the decayed tree and left by the stump. It was assumed that with the time lapse since the moment of decay the properties of the xylem change (MOR and MOE).

## 2. Materials and Methods

The study was conducted in 2019 in one forest with a 95-year-old spruce tree stand at Bialowieża Forest. A large spruce woodland (Figure 1) was chosen for the study, where due to the gradation of the European spruce bark *Ips typographus* beetle the weakening of the trees has been observed for a few years together with successive degradation of the tree stand.

The material for analysing characteristics and properties of the xylem was collected from trees in six groups (Y0 to Y5) and divided based on the time span in which trees decayed. The wood which was analysed did not have any signs of soft rot or hard rot. Y0 indicates healthy trees (control trees) full of vitality, whereas Y5 indicates trees which decayed in the previous 5 years, i.e., in 2014. Y4 indicates trees which decayed 4 years before collecting the sample, i.e., 2015. Y3 were trees decayed in 2016, Y2 in 2017, and Y1 trees decayed in 2018. In each group there were three trees, hence a total of 18 model trees were determined (Table 1). The trees were classified to each group (Y0 to Y5) on the basis of

the data acquired from the inventory and a constant observation of spruce forest conducted by State Forest (Lasy Państwowe) in Hajnówka Forest District. Due to the regular observations of spruce forest decay in Białowieża Forest, obtaining the data concerning the year in which the decay of each tree occurred was relatively simple.

**Table 1.** Basic locality and stand characteristics.

| Group | Tree Number | Height [m] | DBH [cm] | Average Age of Sample Trees | Forest Site Type | GPS (WGS-84) | Mean Precipitation [mm/year] |
|---|---|---|---|---|---|---|---|
| Y0 | 1 | 31.50 | 42.5 | | | | |
| | 2 | 30.60 | 41.5 | | | | |
| | 3 | 29.50 | 39.0 | | | | |
| Y1 | 1 | 32.00 | 43.0 | | | | |
| | 2 | 31.30 | 42.0 | | | | |
| | 3 | 30.20 | 40.5 | | | | |
| Y2 | 1 | 31.60 | 41.5 | | | | |
| | 2 | 30.50 | 40.0 | 95 * | Vaccinio myrtilli-Piceetum | 52°75′39.570″N, 23°75′23.920″E | 593 * |
| | 3 | 29.80 | 38.5 | | | | |
| Y3 | 1 | 32.20 | 43.5 | | | | |
| | 2 | 30.90 | 41.0 | | | | |
| | 3 | 29.80 | 38.5 | | | | |
| Y4 | 1 | 31.50 | 42.5 | | | | |
| | 2 | 30.50 | 41.0 | | | | |
| | 3 | 30.10 | 39.5 | | | | |
| Y5 | 1 | 31.90 | 43.5 | | | | |
| | 2 | 31.20 | 42.0 | | | | |
| | 3 | 30.10 | 40.5 | | | | |

* The data come from the database about forests in Poland.

The material for laboratory analysis was collected from model trees at breast height (DBH) from the arrow in 1.0–1.5 m areas from the butt end as depicted in Figure 2. The manner of collecting the material allowed to avoid the influence of reaction wood in spruce, which usually occurs at the butt end of the arrow.

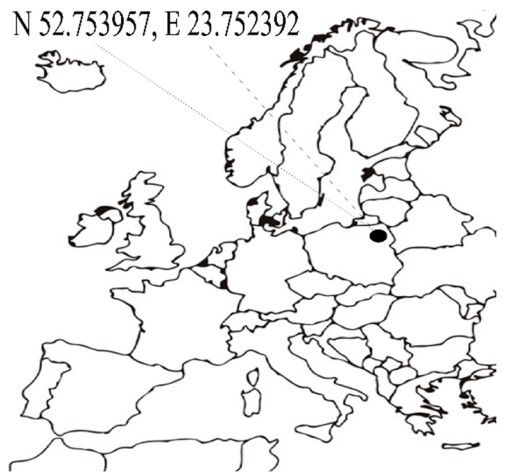

**Figure 1.** Location of the study.

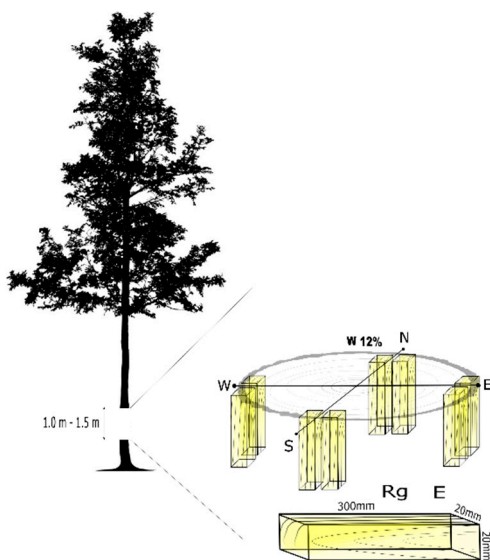

**Figure 2.** The method of collecting samples for the study.

### 2.1. Annual Increment

The analysis of wood ringiness was conducted following PN-D-94021:2013 norms by using Brinell's increment borer. The study was done on solid samples in the available measuring groups. The study was conducted on full cross-sectional area of the samples acquired from the middle parts of the studied constructional elements. For an increased accuracy of the measurement, the analysed measuring length concerned the entire cross-section.

The mean ringiness "S" was measured up to 0.1 mm accuracy based on the dependence:

$$S = l/N \text{ [mm]} \tag{1}$$

where:

S—mean width of annual increment [mm],
N—number of annual rings in the length of the measuring segment [no],
l—the length of the measuring segment [mm],

At the same test stand, the occurrence of earlywood and latewood for further samples has been marked.

### 2.2. Mechanical Properties of Spruce Wood

Measuring mechanical properties of the analysed material was conducted on the samples which were $300 \times 20 \times 20$mm, and it was done using Zwick/Roell Z050 type testing machine together with measuring apparatuses and machines at the Department of Wood-Based Material at the University of Life Sciences in Poznań. The analysis of bending strength MOR (2) and modulus of elasticity MOE (3) at bending was done according to PN-EN 380:1998, PN-EN 408:2004. The analysis of wood properties was carried out at 12% humidity.

In order to determine the 4-point bending strength an apparatus to test the samples in laboratory conditions was used, for that purpose testing machine was used.

While determining bending strength and breaking load at bending, a measurement system in accordance with a 4-point bending strength rule was observed. At the beginning, the sample was subjected to an initial preload, for which the bending value of the arrow was read. Next, the sample was subjected to bending until a total destruction of the sample. Resistance to bending MOR was

determined following the formula (PN-EN 408:2004), and the value of the maximal breaking load was 10 kN.

$$MOR = \frac{aF_{max}}{2W} \tag{2}$$

where:

MOR—resistance to bending [MPa],
a—the distance between the place of loading force and the closest support [m],
$F_{max}$—breaking load [N],
W—indicator of cross-section resistance [m$^3$]; (for a rectangle W = bh$^2$/6, where:
b—width of cross-section of the sample, h—height of cross-section of the sample),

The adopted spacing between the supports was 360 mm, and the distance of pressure source from the support was 80 mm at the initial load 5N. The speed of applied pressure of the loading device was in accordance with PE-EN 408 norm and it did not exceed 0.003 mm/s. The value of bending arrow was determined by using the software of the testing machine.

$$MOE = \frac{\left(l_1 \cdot l_2{}^2\right) \cdot (P_i - P_1)}{4 \cdot (f_i - f_1) \cdot b \cdot h^3} \left[\frac{N}{mm^2}\right] \tag{3}$$

where:

MOE—modulus of elasticity [MPa],
$l_1$—distance between pressure and support [mm],
$l_2$—spacing of supports [mm],
$P_i$—load of the range [N],
$P_1$—initial load [N],
$f_i$—bending arrow at the load [mm],
$f_1$—bending arrow forced by initial load [mm],
b—width of the sample [mm],
h—height of the sample [mm].

Statistical analyses:
At first, the distribution of population was determined following Kolmogorov-Smirnov tests, because the collected data indicated normal distribution at level $p > 0.05$, further analysis was performed by describing a mean standard deviation and post hoc tests. Statistical analyses were conducted adopting STATISTICAS 13 set.
Abbreviations and symbols:

MOR—static bending resistance [MPa]
MOE—modulus of elasticity [MPa]
Y0—sample collected from living trees
Y1—sample collected from trees 1 year after decay
Y2—sample collected from trees 2 years after decay
Y3—sample collected from trees 3 years after decay
Y4—sample collected from trees 4 years after decay
Y5—sample collected from trees 5 years after decay

## 3. Results

The study focused on analysing wood resistance to static bending (MOR) and modulus of elasticity (MOE) of wood of decayed trees, which had decayed in the previous 5 years. Next, the results were compared with the wood of living and fully vital spruces without any signs of disease.

In Table 1 as well as Figures 3 and 4, the results have been collected and presented. They indicate differences of mean values both for static bending (MOR) as well as for modulus of elasticity (MOE) in the comparable groups.

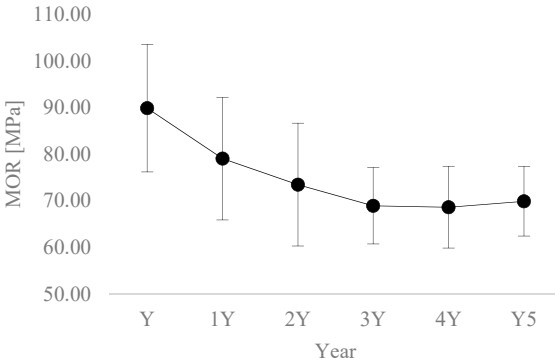

**Figure 3.** Resilience to static bending (MOR) of spruce wood in reference to the time since the tree's decay.

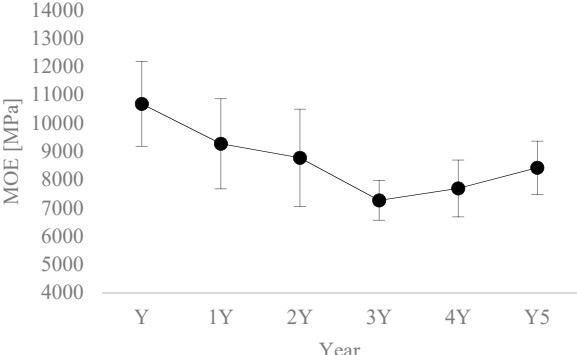

**Figure 4.** Modulus of elasticity (MOE) of spruce wood in reference to the time since the tree's decay.

Higher mean values of MOR (89.85 MPa) and MOE (10,679 MPa) were achieved in the wood of healthy trees and they were statistically significant higher ($p < 0.05$) than the values of the wood derived from trees decayed 3, 4, and 5 years since the samples were collected (Table 2, Figures 3 and 4).

**Table 2.** Statistical properties of resistance to static bending (MOR) and modulus of elasticity (MOE) in Norway spruce wood in reference to time which has elapsed since the tree's decay.

| Variable | Year | Mean | SD | SE | Q25 | Median | Q75 |
|---|---|---|---|---|---|---|---|
| *MOR* [Mpa] | Y0 | **89.85 *** | 13.65 | 4.32 | 83.90 | 93.25 | 96.30 |
| | Y1 | 79.04 | 13.14 | 3.79 | 72.00 | 84.60 | 86.45 |
| | Y2 | 73.46 | 13.16 | 4.39 | 63.00 | 70.80 | 85.60 |
| | Y3 | **68.93 *** | 8.18 | 2.36 | 62.85 | 65.95 | 74.30 |
| | Y4 | **68.63 *** | 8.76 | 2.77 | 65.30 | 69.25 | 74.70 |
| | Y5 | **69.90 *** | 7.48 | 2.25 | 68.20 | 70.40 | 75.80 |
| | *Mean* | 74.85 | 12.91 | 1.61 | 65.70 | 71.35 | 85.10 |
| *MOE* [Mpa] | Y0 | **10,679 *** | 1501 | 475 | 10,686 | 10,941 | 11,269 |
| | Y1 | 9269 | 1594 | 460 | 8002 | 9609 | 10,286 |
| | Y2 | 8772 | 1723 | 574 | 7669 | 8078 | 10,395 |
| | Y3 | **7269 *** | 704 | 203 | 6711 | 7203 | 7854 |
| | Y4 | **7689 *** | 1004 | 317 | 7741 | 7861 | 8068 |
| | Y5 | **8419 *** | 942 | 284 | 8087 | 8735 | 9132 |
| | *Mean* | 8652 | 1662 | 208 | 7675 | 8140 | 9701 |

* Differences statistically significant at the significance level $p < 0.05$ are marked.

The lowest standard deviations of the described properties were noticed in the wood decayed in the 3rd, 4th, and 5th years after collecting the sample and they were for MOR accordingly: Y3 = 8.18 [MPa], Y4 = 8.76 [MPa], Y5 = 7.48 [MPa]; and for MOE: Y3 = 704 [MPA], Y4 = 1004 [MPa], Y5 = 942 [MPa].

Due to the fact that arithmetic mean is not immune to exceptions, for each group the median was calculated separately. The medians of the described properties also require attention. As it can be derived from the data the median of the described properties were close to the arithmetic means and the differences in the range from −2.98 to 5.55 [MPa] for MOR and from −694 to 339 [MPa] for MOE, which in comparison to the values of these properties is relatively not much (Table 2).

A detailed analysis of radial strength distribution (MOR) and modulus of elasticity (MOE) in the tree trunks was conducted. As Figures 5 and 6 present, in the case of sample blank determination (Y0) of the trend is quite standard for coniferous species, i.e., first the mean values of MOR and MOE in the area of the pith are low, then there is an increase and a slight decrease in the peripheral parts of the tree. However, in the xylem of decayed spruces in 3, 4, and 5 years prior to the moment of sample collection, mean variation curves for MOR flattens and remains in the range between 60 and 80 MPa for the entire length of the radius (Figure 5).

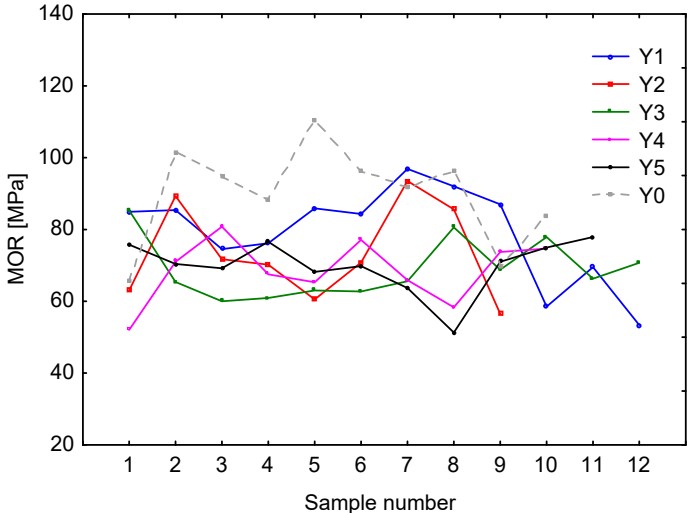

**Figure 5.** The radial variability for static bending resistance (MOR) of spruce wood depending on the period of time since the tree's decay.

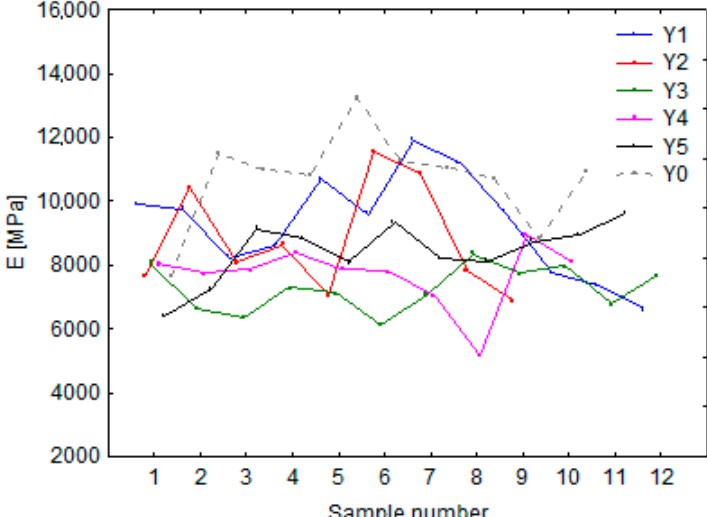

**Figure 6.** The radial variability of modulus of elasticity (MOE) of spruce wood depending on the period of time since the tree's decay.

In the case of modulus of elasticity (MOE) the situation is analogous to the resistance to static bending MOR and the acquired value for wood of trees decayed in 3, 4, and 5 years prior to collecting the samples are in the range 6000–9500 MPa (Figure 6).

Moreover, an analysis of the relationship between resistance to static bending (MOR) and modulus of elasticity (MOE) was conducted, and the achieved results of the correlation coefficients have been presented in Table 3. The statistically significant ($p < 0.05$) correlation coefficients between these wood properties have been noticed between sample blank determination (Y0) and the samples collected from trees in year 1, 2, 3 (Y1, Y2, Y3) after the decay. In year 4 and 5 (Y4, Y5) there were no statistically significant correlations between modulus of elasticity (MOE) and resistance to static bending (MOR).

**Table 3.** Intra-group correlations between MOR and MOE.

| | | MOR | | | | | |
|---|---|---|---|---|---|---|---|
| | | **Y1** | **Y2** | **Y3** | **Y4** | **Y5** | **Y0** |
| **MOE** | Y1 | 0.942974 | | | | | |
| | Y2 | | 0.961817 | | | | |
| | Y3 | | | 0.625188 | | | |
| | Y4 | | | | 0.408902 | | |
| | Y5 | | | | | 0.144510 | |
| | Y0 | | | | | | 0.950618 |

Correlation coefficients are statistically significant at the level of $p < 0.05$.

## 4. Discussion

Forest damages caused by winds and also heavy snow or ice—sleet is an important ecological aspect. It can have serious economic consequence due to limitations in wood production, and at the same time, social and ecological ones.

Polish Forests are in the range of extreme abiotic factors, such as strong winds, intense precipitations of wet snow, rime ice, or drought. All these phenomena have particularly intensified this century. In the weakened tree stands there can be an exponential development of infectious diseases and gradation of insects. Such tree stands become less stable and hence greater sustainability to new damages [28].

The literature of the subject describes various risk models caused by wind which affects single trees as well as entire tree stands [29–36]. Current risk prediction models for damages caused by winds are often focused only on "medium" trees within homogenous tree stands [37] and refer to healthy trees without considering tree stands in which lesions and disease symptoms have occurred and the tree stands decayed as a result. Due to their legal status these specimens have not been removed as they were located in nature reserves or parks which have a significant ecological and social role.

There are also numerous academic papers focused on risk management in order to minimize the damages in forests caused as a result of various factors including wind or the changing climate that affect the risks [38–40]. However, the results of the research refer also to entire tree stands or stocking in urban areas and generally refer to healthy trees, without any visible symptoms of a disease.

The paper does not propose a model supporting decayed tree stand management which are vulnerable to damages. However, the paper does present results of the research referring to mechanical stability of spruce tree stands and potential hazard they can pose, and which decayed as a result of weakening process and gradation of European spruce bark beetle in Białowieża Forest.

As a consequence of the conducted research, key mechanical wood properties of trees in various periods after the decay, significant due to their biomechanical stability, were determined. The trees in question were spruces which decayed in yearly periods between 2014 and 2019.

Ancelin et al. [41] emphasizes that the models predicting wind resistance for tree stands are usually based on calculating the critical wind speed above which a medium tree is broken or uprooted. Such an approach does not apply to all tree stands, as in each tree stand the wind affects trees

differently and the trees indicate different critical wind resistance, which is connected with a natural variability of wood properties [42], the morphological tree architecture [20,43], industrial, and economic decisions [44], and properties of the entire tree stand which are considered in there referred works which research and study risk models.

The models rarely consider the variable resistance of the xylem or the wood defects in the tree, and they actually underestimate (belittle) or overlook them. The properties of the xylem or the defects, and above all its mechanical properties, reflect on the tree stability and seem a crucial factor in predicting damages to occur which could be life-threatening and hazardous to humans and causing significant economic losses.

The static bending resistance of spruce wood varies, and depends on many factors, and ranges from 49 up to 136 [MPa]; however, the modulus of elasticity at static bending of spruce wood is between 7300 even up to 21,000 [MPa]. The achieved results for the wood of healthy trees are a little above the average ascribed to static bending strength (MOR); however, static bending strength of decayed trees is below the average for this range. The results for Young's modulus (E) are in the lower end of the range, particularly for the wood of the decayed trees.

The research indicates the existence already, in the 3rd year after the decay, of a lowered resistance to static bending (MOR) and modulus of elasticity (MOE) which are statistically significant ($p < 0.05$). The lowered resistance to static bending in the 3rd year after decay was by over 23% lower in comparison with living trees (Y0). Similarly, modulus of elasticity in the 3rd year after the decay was by 32% lower than in the wood of living trees (Y0).

As it is known, wood properties of coniferous species are shaped by the properties of the xylem as well as the shape of the latewood in the annual growth ring [45].

The relationship between the annual growth with density and wood properties of coniferous species have been described by many scholars [46,47] and it is assumed that the narrower the annual ring, the higher density and wood properties, and vice versa. Moreover, it is considered that with the increasing share of this wood, i.e., from the pith to the girth, the density and properties of the xylem gradually increase [12,48,49]. In the analysed material this rule applied only to wood derived from the living trees (Y0) and in the 1st and 2nd year after decay (Y1 and Y2); however, in the xylem of trees decayed 3, 4, and 5 years (Y3, Y4, and Y5) after collecting the sample there was a significant flattening of the trend on radius both in MOR and MOE. Although the trees chosen for sample collection did not have any symptoms of hard or soft rot and were deprived of any other defects, hence it can be assumed that after the physiological decay, the changes at ultrastructural level of the wood, such as depolymerisation of the cellulose in tracheid cell wall occurred and which had an impact on its properties.

In order to exclude the influence of the xylem properties on the achieved results, the analysis of radial variation of annual ring width and particular annual ring zones of (earlywood, latewood) was conducted. In the analysed groups there were no significant differences in the latewood annual ring width. It has been observed that the annual rings in the trees from Y1 were on average wider (Figure 7). The annual ring increment concerned in this case earlywood zone, which has conductive functions and not mechanicals ones, hence from mechanical stability viewpoint of the tree it did not have a significant impact.

An unusual phenomenon is also a lack of significant relations between static bending resistance and modulus of elasticity (MOE) in the wood of decayed trees 4 and 5 years (Y4, Y5) before collecting samples (Table 3). Similarly to the unusual distribution of MOR and MOE on the radius for spruce, also in this case it should be assumed that the form and cellulose structure, which determines mechanical properties of xylem, play a significant role [50,51].

The results, to a certain degree define often an overlooked role of the xylem properties in creating various models referring to both the risk of tree breaking and risk management. Furthermore, greater risk of an accident with such trees is generated and hence it justifiable to acknowledge them as particularly hazardous. At the same time, it should be kept in mind that there is a significant

and natural variation of the xylem within a single species and very often there is an overlap of various factors influencing wood formation. Numerous conclusions can be reached concerning the role of mechanical wood properties in shaping tree stability in different life stages as well as after their decay. Hence, the described properties are a significant element which should be taken into consideration in predicting damages in forests and accidents involving trees.

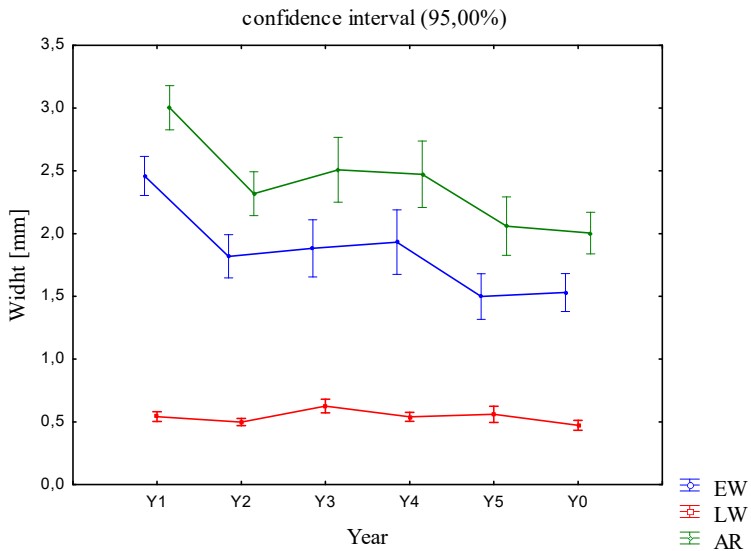

**Figure 7.** Mean width of annual rings in comparable tree groups (EW—early wood; LW—latewood; AR—annual ring).

## 5. Conclusions

The results of the study signify the progressive changes of wood properties in decayed spruces left by the stump. The probable changes in the ultrastructure of the xylem significantly decrease wood properties. Depending on the function played by the forests, where the tree stands have been weakened with gradation and tree decay, it is possible to indicate two types of action:

- In forests which play a social function, in nature reserves and parks, spruces can be considered potentially hazardous after the 3rd year since their decay and the areas with such trees should be excluded from recreational purposes (touristic purposes).
- In industry tree stands, the areas where dead trees have been decaying for 3 years and more should be excluded from use. Furthermore, in the 1st and 2nd year after the decay, spruce wood can be to some extent used in industry. However, it cannot be used as construction wood, but as fuel wood or left for further decay until total decomposition (dead tree).

**Author Contributions:** T.J. and K.K. planned and designed the research; T.J., K.K. and A.T. performed the experiments; J.K., M.A.-J., M.W. collected and analyzed the data, T.J. performed the statistical analysis; T.J., K.K., W.G. and M.A.-J. wrote the manuscript; T.J., K.K., A.T. revised the manuscript. All authors have read and agreed to the published version of the manuscript.

**Funding:** The publication is co-financed within the framework of Ministry of Science and Higher Education programme as "Regional Initiative Excellence" in years 2019–2022, project number 005/RID/2018/19

**Conflicts of Interest:** The authors declare no conflict of interest.

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
