# Peer review of "Influence of the Tree Decay Duration on Mechanical Stability of Norway Spruce Wood (Picea abies (L.) Karst.)"

_forests, doi:10.3390/f11090980_

Round 1
Reviewer 1 Report
Interesting article on an actual theme. I have just few comments or questions:
- According to EN 408 - the span between supports for four point bending test is given as 18.h. For the test sample with hight of 20 mm, it should be 360 mm (not 240 mm).
- Considering the wind load, which is also mentioned in this article, whrere there any differencies in the width of anual rings, MOR and MOE depending on the cardial points of taken test samples?
- Test samples for Y1 - Y5 were taken from parts with detected decay or the were without decay (taken from tha tree area without decay)? it is hard to understand it from the text.
- Figure 6 - "semple" correct to "sample"
Author Response
Dear Reviewer,
In the table below I have collected and referred to the reviewer comments and our responses to them. We practically agree with all these observations and comments and we have already made the necessary corrections and changes in the text. We would like to thank the reviewers for their valuable comments and advice.

Reviewer 2 Report
Study presents interesting topic related also to modern global warming treats. Similar problems (decaying spruce forests) are reported all over Europe so wood quality studies are important.
All my comments are listed in attached manuscript.

Author Response

(The authors gave the same response as above.)
